# Exploring Pro-Inflammatory Immunological Mediators: Unraveling the Mechanisms of Neuroinflammation in Lysosomal Storage Diseases

**DOI:** 10.3390/biomedicines11041067

**Published:** 2023-04-01

**Authors:** Manoj Kumar Pandey

**Affiliations:** 1Cincinnati Children’s Hospital Medical Center, Division of Human Genetics, Cincinnati, OH 45229-3026, USA; manoj.pandey@cchmc.org; 2Department of Pediatrics, University of Cincinnati College of Medicine, Cincinnati, OH 45267-0515, USA

**Keywords:** mental health, neuroinflammation, accumulation of toxic substances, immune checkpoint pathways, therapeutic targets

## Abstract

Lysosomal storage diseases are a group of rare and ultra-rare genetic disorders caused by defects in specific genes that result in the accumulation of toxic substances in the lysosome. This excess accumulation of such cellular materials stimulates the activation of immune and neurological cells, leading to neuroinflammation and neurodegeneration in the central and peripheral nervous systems. Examples of lysosomal storage diseases include Gaucher, Fabry, Tay–Sachs, Sandhoff, and Wolman diseases. These diseases are characterized by the accumulation of various substrates, such as glucosylceramide, globotriaosylceramide, ganglioside GM2, sphingomyelin, ceramide, and triglycerides, in the affected cells. The resulting pro-inflammatory environment leads to the generation of pro-inflammatory cytokines, chemokines, growth factors, and several components of complement cascades, which contribute to the progressive neurodegeneration seen in these diseases. In this study, we provide an overview of the genetic defects associated with lysosomal storage diseases and their impact on the induction of neuro-immune inflammation. By understanding the underlying mechanisms behind these diseases, we aim to provide new insights into potential biomarkers and therapeutic targets for monitoring and managing the severity of these diseases. In conclusion, lysosomal storage diseases present a complex challenge for patients and clinicians, but this study offers a comprehensive overview of the impact of these diseases on the central and peripheral nervous systems and provides a foundation for further research into potential treatments.

## 1. Introduction

Lysosomal storage diseases are characterized by a deficiency in lysosomal enzymes that are responsible for breaking down and recycling cellular waste. This enzyme deficiency leads to the accumulation of toxic substances in the lysosomes, causing cellular damage and dysfunction, particularly in the nervous system [1]. This damage, in turn, drives the development of neuroinflammation, leading to the progression of symptoms and functional decline in lysosomal storage diseases [2,3,4]. The genetic defects underlying lysosomal storage diseases result in the excessive production of specific substrates, including glucosylceramide (GC), globotriaosylceramide (Gb3), globotriaosylsphingosine (lyso-Gb3), ganglioside GM1 (GM1), ganglioside GM2 (GM2), sphingosine (Sph), glycosphingolipids (GlycSph), sphingomyelin (Sm), cholesterol (Ch), ceramide (Cer), galactosylceramide (GalCer), galactosylsphingosine (GalSph), cholesteryl esters (CEs), and triglycerides (TGs), as listed in Table 1. These substrates drive the disease process in several lysosomal storage diseases, including Gaucher disease, Fabry disease, GM1 gangliosidosis, GM2 gangliosidosis, Tay–Sachs disease, Sandhoff disease, Niemann–Pick type C disease, Farber disease, Krabbe disease, and Wolman disease (Table 1).

The excessive accumulation of substrates in lysosomal storage diseases has been shown to have significant impacts on several types of cells, including microglial cells, astrocytes, oligodendrocytes, neurons, Schwann cells, monocytes (MO), macrophages (Mϕs), dendritic cells (DCs), natural killer (NK), T, and B cells [33,34,35,36,37,38,39]. These cells respond to the substrate accumulation by generating an enormous amount of pro-inflammatory cytokines, including interferon gamma (IFNγ), tumor necrosis factor alpha (TNFα), interleukin 1 beta (IL1β), IL6, IL17, C-C motif ligand 2 (CCL2), CCL4, CCL5, C-X-C motif ligand 10 (CXCL 10), complement proteins such as complement component 1q (C1q), C3, and C5a, and autoantibodies to certain glycolipids [5,36,40,41,42,43]. This massive generation of pro-inflammatory cytokines, complement proteins, and autoantibodies has been linked to cellular damage and dysfunction, particularly in the central and peripheral nervous systems, leading to the development of neuroinflammation and the progression of disease symptoms in lysosomal storage disease [5,36,40,41,42,43].

There are several pro-inflammatory mediators, including IFNγ, IL1β, TNFα, IL6, IL17, CCL2, CCL4, CCL5, CXCL10, autoantibodies, and complement proteins, that have shown to cause damage to the blood–brain barrier and the blood–nerve barrier, which are essential for maintaining the homeostasis of the central and peripheral nervous systems [44,45,46]. The blood–brain barrier is a highly selective semipermeable barrier that separates the circulating blood from the brain’s extracellular fluid. It regulates the entry of substances, such as nutrients and waste products, into the brain and prevents the entry of potentially harmful substances, including pathogens and toxins. Disruption of the blood–brain barrier can lead to the entry of inflammatory cells and mediators into the brain, which can contribute to neuroinflammation and neurological dysfunction [47,48,49]. Similarly, the blood–nerve barrier is a specialized barrier that separates the peripheral nerve tissue from the circulating blood. It plays a crucial role in maintaining the extracellular environment of the peripheral nerve tissue, regulating the entry of nutrients and waste products, and preventing the entry of harmful substances. Damage to the blood–nerve barrier can lead to the entry of pro-inflammatory mediators and immune cells into the peripheral nerve tissue, which can cause peripheral neuropathy and other nerve-related disorders [50,51,52,53].

The accumulation of substrates in lysosomal storage diseases has a far-reaching impact and can profoundly affect the immune system and neurological function of the body. This excess accumulation can generate pro-inflammatory cytokines, chemokines, complement proteins, and autoantibodies, which could contribute to the development of neuroinflammation and the progression of disease symptoms by affecting the blood–brain barrier and blood–nerve barrier. Understanding these disease mechanisms is crucial for identifying potential biomarkers, accurately diagnosing disease stages, targeting specific areas for treatment, and developing clinical trials aimed at curing neuroinflammation and neurodegeneration in lysosomal storage diseases.

## 2. Gaucher-Disease-Associated Neuroinflammation: Deciphering the Complex Interactions between Neurological and Immune Systems 

Gaucher disease is a rare, autosomal recessive genetic disorder, which affects 1/40,000 to 1/60,000 live births [43,54,55]. Gaucher disease occurs due to the mutations in *GBA1* (in human)/*Gba1* (in mouse) which leads to the functional disruption of the enzyme, Acid β-glucosidase (GCase; EC 4.2.1.25) responsible for the biochemical degradation of GC and GS that are essential for the proper function of skin, cell survival, and embryonal development [56,57,58,59]. The genetic deficiency of GCase and consequent excess Mϕs and microglial cell accumulation of GC and GS affect both visceral organs and the brain in Gaucher disease [36,60]. 

Gaucher disease can be classified into three types. Type 1 primarily affects the internal organs, such as the liver, spleen, lung, lymph node, bone, and kidney. The symptoms associated with type 1 Gaucher disease include anemia, thrombocytopenia, hyper-gammaglobulinemia, splenomegaly, hepatomegaly, skeletal weakness, B cell malignancies, and mild brain inflammation [42,60,61]. 

On the other hand, type 2 and type 3 Gaucher diseases mainly affect the central nervous system tissues, leading to the development of neurological symptoms. These symptoms may include myoclonus with selective dentate abnormalities, generalized epilepsy, seizures, slowing of horizontal saccadic eye movements, ataxia, spasticity, oculomotor abnormalities, hypertonia of neck muscles, extreme arching of the neck, bulbar signs, limb rigidity, occasional choreoathetoid movements, and progressive dementia [62,63,64,65,66,67,68,69,70,71,72,73].

The Gba1-prone mouse model of type 1 Gaucher disease, specifically the D409V/null; 9V/null; Gba1^9V/−^ strain, has provided valuable insights into the pathophysiology of this disorder. This model has shown an excessive accumulation of GC in various tissues, including the liver, spleen, lymph nodes, and lungs(5,7,36,61). Furthermore, elevated levels of GC-specific immunoglobulin G (IgG) autoantibodies have been observed in the circulatory system autoantibodies [36,74].

In addition, this mouse model has demonstrated a significant increase in the generation of pro-inflammatory cytokines, such as IFNγ, TNFα, IL1β, IL6, IL12, IL17, IL21, and IL23, as well as chemokines, including CCL2, CCL3, CCL4, CCL5, CCL6, CCL9, CCL17, CCL18, CCL22, CXCL1, CXCL2, CXCL8, CXCL9, CXCL10, CXCL11, CXCL12, and CXCL13 (5,6,7,36). The research has also revealed that the Gba1^9V/−^ mice mouse model of Gaucher type 1 disease exhibits elevated levels of growth factors, such as transforming growth factor-beta 1 (TGFβ1), hepatocyte growth factor (HGF), Mϕ-colony-stimulating factor (MCSF), granulocyte-colony-stimulating factor (GCSF), and granulocyte-Mϕ colony-stimulating factor (GMCSF), as well as an increase in the recruitment of diverse immune cells, such as DCs, Mϕs, T cells, and B cells [6,7,36]. These results provide us with a more in-depth comprehension of the immunological transformations that transpire in type 1 Gaucher disease. Moreover, these studies also point towards several potential biomarkers and targets for curative interventions. Furthermore, Gba1^9V/−^ mice showed higher lung expression of activating Fc gamma (Fcγ) receptors (e.g., FcγRI, FcγRIII, and FcγRIV), inhibitory FcγR (e.g., FcγRIIB), C1q subcomponent, alpha polypeptide (C1qa), C-type lectin domain family 4 member n (Clec4n), C-type lectin domain family 5, member a (Clec5a), C3a receptor1 (C3aR1), and C5a receptor1 (C5aR1), as well as increased liver expression of C-type lectin domain family 7, member a (Clec7a) [75]. Mϕs and DCs of the Gba1 ^9V/−^ mouse model of type 1 Gaucher disease have shown higher expression of C5a and C5aR1 [36]. Liver, spleen, and lung and these tissues-derived CD4+ T cells that were stimulated with anti-CD3 and CD28, as well as GC-stimulated liver, spleen, and lung-derived DCs and CD4+ T cells, were found to produce a significant amount of pro-inflammatory cytokines. These cytokines include IFNγ, TNFα, IL1β, IL6, IL12, IL17, IL21, and IL23 in Gba1 ^9V/−^ mouse model of type 1 Gaucher disease [6,7,36,74,75,76,77,78,79,80]. 

Type 1 Gaucher disease patients have been observed to have elevated levels of GC-specific IgG immune complexes (GC-ICs) in their serum and/or plasma [8,47]. In a model of human Gaucher disease using conduritol B epoxide (CBE)-induced GCase-targeted Mϕs stimulated with GC-ICs, there was an increase in the production of pro-inflammatory cytokines, including IFNγ, TNFα, IL1β, IL6, IL12, IL17, IL21, and IL23 [5,7,76]. In both type 1 Gaucher disease mouse models and human patients with Gaucher disease, GC-induced innate and adaptive immune inflammation could impact the brain and cause neurological abnormalities. However, further studies are needed to confirm this.

Type 2 and type 3 Gaucher diseases are the acute and chronic neurological forms of the disease, also referred to as neuronopathic Gaucher disease [81,82,83]. In these forms, there is an excess accumulation of GC in the brain, which impacts several regions, such as the substantia nigra reticulata, the reticulotegmental nucleus of the pons, the cochlear nucleus, and the somatosensory system. This leads to MO differentiation into microglial cells, proliferation and activation of microglial cells and astrocytes, and massive generation of pro-inflammatory cytokines (IFNγ, TNFα, IL1α, IL1β, and IL6), chemokines (CCL2, CCL3, and CCL5), reactive oxygen species (ROS), nitric oxide (NO), and growth factors (MCSF and TGFβ). Additionally, downregulation of brain-derived neurotrophic factor (BDNF) and nerve growth factor (NGF) has been observed, which collectively leads to neuron loss and early death. These effects have been observed in several Gba1 prone models, such as Gbaflox/flox; Nestin-Cre mice, K14-lnl/lnl mice, 4L; C*, C57BL/6J-Gbatm1Nsb, as well as Conduritol B epoxide-induced chemical model of neuronopathic Gaucher disease [78,84,85,86,87,88,89,90,91,92,93,94]. 

Patients with neuronopathic Gaucher disease exhibit brain accumulation of cytotoxic glycosphingolipids, which leads to microglial cell activation and upregulation of pro-inflammatory cytokines, such as TNFα, IL1β, and IL6. These events have been linked to the development of severe and chronic brain inflammation, resulting in the loss of neurons and early death [78,92,93,95,96,97,98,99,100]. Recent studies using induced pluripotent stem cell (iPSC)-derived mϕs and lymph nodes from patients with type 2 and type 3 Gaucher diseases have demonstrated that the C5a-C5aR1 axis triggers the induction of TNFα and TGFβ signaling [101,102]. The observed events, such as the induction of TNFα and TGFβ signaling triggered by the C5a-C5aR1 axis, have the potential to cause significant harm to patients with neuronopathic Gaucher disease. Inflammation in tissues and organ failure can have a profound impact on the body and can lead to significant morbidity and even mortality. As such, further research on the mechanisms underlying these events is critical in the development of effective treatments and therapies for patients with neuronopathic Gaucher disease. 

Several of these studies suggest that mutations in the GBA1/Gba1 gene lead to the accumulation of glucocerebroside (GC) in the brain, triggering the activation of microglial cells and resulting in excessive generation of pro-inflammatory cytokines, including IL1β, TNFα, and IL6. Similarly, in peripheral organs such as the liver, spleen, and lung, the excess accumulation of GC leads to the recruitment and activation of immune cells, overproduction of IgG, IgM, and IgA autoantibodies to various protein and glycolipid antigens, and the overproduction of pro-inflammatory cytokines, chemokines, growth factors, and complement cascade proteins (e.g., C3, C3a, C5, and C5a). The complex interactions between such genetic mutations and a cascade of cellular processes and immune system activation in both the central nervous system and peripheral organs can affect the blood–brain barrier, potentially leading to brain inflammation and neuronal cell death in Gaucher disease. These findings provide insight into the disease mechanism and may offer new avenues for research and treatment.

## 3. Fabry-Disease-Associated Neuroinflammation: Deciphering the Complex Interactions between Neurological and Immune Systems 

Fabry disease is an X-linked recessive inborn error of glycosphingolipid metabolism, which affects 1/40,000–1/117,000 live births [103,104]. Fabry disease occurs due to *GLA* (in human)/*Gla* (in mouse) defect and the resultant deficiency of the lysosomal enzyme α Gal A (E.C. 3.2.1.22) and the excess accumulation of glycosphingolipids, mainly Gb3 and lyso-Gb3 in the affected cells and body fluids [8,9,10,11,12,13,14,15,16,105]. Gb3 and lyso-Gb3 are critical for controlling cellular activation, cell proliferation, and microbial growth [106,107,108,109]. However, *GLA*/*Gla* defect and the resultant excess accumulation of Gb3 and lyso-Gb3 affect blood vessel walls, renal epithelial cells, endothelial cells, pericytes, podocytes, tubular cells of the distal tubule and loop of Henle, vascular smooth muscle cells, cardiomyocytes, and neurons of the peripheral and autonomic nervous system and develop several of the disease symptoms, i.e., heart enlargement, renal failure, stroke, gastrointestinal difficulties, decreased sweating, fever, angiokeratomas, and burning pain in the arms and legs [8,9,10,11,12,13,14,15,16,107,110,111]. 

The plasma and or renal biopsy samples from the mouse model of Fabry disease showed higher levels of Gb3, complement components (e.g., C3 and iC3b) [112], and IFNγ [76,112]. Similarly, excess accumulation of Gb3 has been observed in plasma, peripheral blood mononuclear cells (PBMCs), conjunctival biopsies, central nervous system tissues, and neurons of the autonomic and peripheral nervous system of patients with Fabry disease [112,113,114,115]. Elevated levels of several components of the complement system, i.e., C1qc, C3, iC3b, C4, and C4b, complement factor B precursors (C3/C5 convertase) have been observed in the serum, plasma, and brain of patients with Fabry disease [14,112]. Serum, plasma, PBMCs, and several of the immune cells that include MOs, DCs, and NK cells have shown massive production of pro-inflammatory cytokines (e.g., IFNγ, TNF α, IL1β, and IL6) in patients with Fabry disease [11,76,113]. The progressive accumulation of Gb3 in the neurons of the autonomic and peripheral nervous system has been linked to the increased production of pro-inflammatory cytokines and oxidative stress and development of the cerebrovascular complications, organ failure, and premature death in Fabry disease [8,9,10,11,12,13,14,15]. 

The results of this study indicate that mutations in the *GLA*/*Gla* gene lead to the accumulation of Gb3/Lyso-Gb3 in central nervous system tissues, triggering the activation of microglial cells and resulting in increased production of pro-inflammatory cytokines, such as IFNγ, TNFα, IL1β, and IL6, in Fabry disease. In peripheral organs, such as the liver, spleen, lung, kidney, and heart, the accumulation of Gb3/Lyso-Gb3 leads to massive activation and recruitment of immune cells, including NK cells, MOs, and DCs. The effector function of these immune cells results in the excess production of pro-inflammatory cytokines and complement proteins, which contribute to organ damage in Fabry disease. The development of such a pro-inflammatory environment in the central nervous system and peripheral organs can damage the blood–brain barrier and lead to neuronal cell death. These findings can contribute to a better understanding of the disease processes and the development of effective therapies to improve the prognosis of patients with Fabry disease.

## 4. GM1-Gangliosidosis-Associated Neuroinflammation: Deciphering the Complex Interactions between Neurological and Immune Systems 

GM1 gangliosidosis is an inherited neurodegenerative disorder, which affects 1 in 100,000–200,000 live births [116]. GM1 gangliosidosis occurs due to the *GLB1 (in human)/Glb1 (in mouse)* gene defects, which is critical for encoding an enzyme β gal (E.C. 3.2.1.23) responsible for the hydrolysis of the GM1, oligosaccharides, and keratan sulfate, all of which are critical for performing many of the cellular signaling and the proper functions of the brain and visceral organs [117,118]. The *GLB1*/*Glb1* defects and the resultant deficiency of β gal cause the excessive accumulation of GM1 in the central nervous system tissues and lead to the development of severe progressive neurological deficits, whereas the excess accumulation of the keratan sulfate and oligosaccharide in peripheral organs (e.g., liver, spleen, heart, and bone) develops the systemic sickness [17,18,19]. GM1 gangliosidosis has been classified into three clinical forms, i.e., type 1, type 2, and type 3. Type 1 GM1 gangliosidosis is an infantile form of the disease that involves rapid cognitive decline, visceromegaly, skeletal abnormalities, and death within 2 years of age. Type 2 GM1 gangliosidosis is the late-infantile or juvenile form of the disease, which exhibits additional wide-ranging phenotypes, such as slow systematic neurological defects and less skeletal vicissitudes. Type 3 GM1 gangliosidosis is a lesser adult and chronic phenotype, which appears with moderate cognitive deterioration and progressive ataxia [119,120]. Cerebral cortex neurons, astrocytes, and microglial cells of Glb1^−/−^ and Glb1^+/−^ mouse models of GM1 gangliosidosis have shown excess accumulation of GM1 [121]. The brains of Glb1^−/−^ and Glb1^+/−^ mouse models of GM1 gangliosidosis have shown activated subsets of Mϕs and microglial cells and the massive production of IL1β and TNFα [122,123]. 

The neurons, oligodendrocytes, fibroblasts, and iPSCs from patients with GM1 gangliosidosis have shown excess accumulation of GM1 [19]. The neuroprogenator cells (NPSCs) from patients with GM1 gangliosidosis have shown higher levels of TNF α, IL1β, and IL6, Additionally, *GLB1* gene defects and the resultant excess accumulation of GM1 have been linked to the microglial cells activation, excess generation of pro-inflammatory cytokines, and chemokines that lead to the neurodegeneration, motor function defects, and the hind limb paralysis in GM1 gangliosidosis [121,122,124]. However, the exact mechanisms by which GM1 propagates the neuroinflammation in GM1 gangliosidosis are still unclear. Studies have observed IgG antibodies against GM1 and their link to acute motor axonal, acute motor–sensory axonal, and acute sensory ataxic neuropathies [125,126,127,128,129,130]. Furthermore, IgG antibodies against GM1-injected animal models have shown the involvement of complement activation in the induction of motor and sensory nerve damage [131,132]. 

The findings of this study suggest that defects in the *GLB1*/*Glb1* gene cause the accumulation of GM1 in the central nervous system, leading to the activation of microglial cells and excessive generation of pro-inflammatory cytokines, such as TNFα, IL1β, and IL6. This pro-inflammatory environment can contribute to neuronal cell death in GM1 gangliosidosis. In peripheral organs, such as the liver, spleen, heart, and bone, the accumulation of keratan sulfate and oligosaccharides results in the overproduction of pro-inflammatory cytokines, including TNFα, IL1β, and IL6, which lead to damage to the visceral tissues. These results provide valuable insight into the underlying mechanisms of GM1-induced inflammation in both the central nervous system and peripheral tissues. This information may serve as a basis for developing more effective treatments and therapies for these conditions in the future.

## 5. GM2-Gangliosidosis-Associated Neuroinflammation: Deciphering the Complex Interactions between Neurological and Immune Systems

Tay–Sachs and Sandhoff diseases are classical examples of GM2 gangliosidosis [133]. The worldwide prevalence of Tay–Sachs disease is ~1 in 320,000 and Sandhoff disease is ~1 in 1,000,000 live births [134]. Sandhoff disease occurs due to mutations in the *HEXB* (in human)/*Hexb* (in mouse) gene and the resulting deficiency of the β subunit of the enzyme hexosaminidase and the resulting excess central nervous system tissue accumulation of GM2, whereas Tay–Sachs disease occurs due to mutations in the *HEXA* (in human)/*Hexa* (in mouse) gene and the resulting deficiency of the *α* subunit of the hexosaminidase and the resulting excess central nervous system tissue accumulation of GM2 [24]. The GM2 is a glycosphingolipid (composed of a ceramide linked to oligosaccharide and specific numbers of sialic acids), which plays an integral role in cell signaling and metabolism [135,136]. However, the genetic deficiency of the α/β hexosaminidase in Tay–Sachs/Sandhoff disease leads to microglial cell activation, increased production of IL1α, IL1β, and TNFα in the brain, as well as neurodegeneration and death at the age of 3–5 years [137].

Sandhoff disease affects both the central nervous system and visceral organs and presents with a range of symptoms, including but not limited to hepatosplenomegaly, motor weakness, early-onset blindness, spasticity, myoclonic seizures, macrocephaly, cherry-red spots in the eye, recurrent respiratory infections, heart murmurs, and doll-like facial features [20,21,22,23,24,25]. In contrast, Tay–Sachs disease primarily affects the central nervous system and is characterized by symptoms, such as dementia, reduced eye contact, heightened startle response to noise, deafness, difficulty swallowing, blindness, cherry-red spots in the retina, paralysis, seizures, and progressive loss of mental capacity [20,21,22,23]. The exact mechanisms by which excess tissue accumulation of GM2 triggers neuroinflammation and neurodegeneration in Sandhoff and Tay–Sachs diseases are unclear. 

Sera and central nervous system neurons from mouse models of Sandhoff disease (*Hexb*^−/−^) and Tay–Sachs disease (*Hexa*^−/−^) have shown increased central nervous system levels of GM2 as well as the IgG antibodies to GM2 [24,138]. The spinal cords of *Hexb*^−/−^ mice have shown elevated levels of IgG, low-affinity FcγRIII, and C1qc [23]. *Hexb*^−/−^ mice brain regions (e.g., parenchyma, brainstem, thalamus, and spinal cord) have shown increased levels of the F4/80^+^ CD68^+^, CD11b^+^ Gr1^+^ CD68^+^ Iba1^+^ CD68^+^ subsets of microglial cells and Mϕs [23,24]. The brain tissue and the neurological cells (e.g., microglial cells and astrocytes) of *Hexb*^−/−^ mice have shown increased production of TNF α, IL1β, and TGFβ1 and CCL3 [23,139]. Thalamus, brainstem, cortices, and spinal cord of these mouse models of GM2 gangliosidosis have also shown neuron loss and strogliosis [21,24,140]. Similarly, human patients with such GM2 gangliosidosis exhibited excess brain accumulation of GM2, microglial cell activation, upregulation of the genes responsible for the production of complement proteins (e.g., C1q, C3, and C4b), and increased production of pro-inflammatory cytokines (e.g., IL1α, IL1β, and TNFα), as well as the neurodegeneration and death at the age of 3–5 years [20,21,22,23,24,25,141]. 

The results of this study suggest that defects in the *HEXB*/*Hexb* and *HEXA*/*Hexa* genes lead to the accumulation of GM2 in the brain and spinal cord, which can activate microglial and/or neuronal cells and result in the overproduction of pro-inflammatory cytokines, such as TNFα, IL1β, and IL6. This pro-inflammatory environment can cause damage to the blood–brain barrier, leading to neuronal cell death and cognitive defects in GM2 gangliosidosis. In addition to the effects in the central nervous system, peripheral organs can also be affected by the accumulation of GM2. The infiltration of immune cells and the massive generation of complement proteins can lead to an excessive inflammatory response, further exacerbating the damage to tissues and organs. The complex interactions between genetic mutations and the resulting cascade of cellular processes and immune system activation contribute to the development of a pro-inflammatory environment that affects both the central nervous system and peripheral organs in GM2 gangliosidosis. Understanding the underlying mechanisms of the pro-inflammatory environment in GM2 gangliosidosis is critical for the development of effective treatments and therapies. By deepening our knowledge of the disease processes, studies can identify new targets for intervention and improve the prognosis of patients with GM2 gangliosidosis.

## 6. Niemann–Pick Type C-Disease-Associated Neuroinflammation: Deciphering the Complex Interactions between Neurological and Immune Systems

Niemann–Pick type C (NPC) disease is a rare progressive genetic disorder that affects ~1 in 100,000–120,000 live births [26,142]. NPC disease is caused by mutations in *NPC1* and *NPC2* (in human)/*Npc1* and *Npc2* (in mouse) genes [143,144]. *NPC1/Npc1* encodes a large transmembrane protein of the late endosome/lysosome known as Niemann–Pick C1 protein (NPC1), which help trafficking of Ch and GlycSph from lysosomes to other membrane compartments, including the endoplasmic reticulum, plasma membrane, trans-golgi network, and mitochondria, to meet their Ch/GlycSph requirements [145,146,147]. Similarly, *NPC2/Npc2* encodes a soluble lysosomal Ch-binding protein termed as NPC2 protein, which transports Ch and GlycSph from inner lysosomal vesicles to the limiting membrane of the lysosome [148,149,150]. However, *NPC1/Npc1* and *NPC2/Npc2* defects lead to the excess accumulation of Ch, Sph, GlycSph, and Sm in the lysosomes of both brain and visceral organs (e.g., liver, spleen, and lung), whereas *NPC1/Npc1* defects account for ~95% of all clinical cases and *NPC2/Npc2* defects are accountable for the remainder [151,152,153,154]. It is interesting to note that there are some similarities that exist between *NPC1/2* and *NPC A/B*, which is also known as acid sphingomyelinase deficiency (ASMD) that affects ~1 in 250,000 [155]. The deficiency of the sphingomyelinase and the resulting excess tissue accumulation of the Sm and chitinase-like protein Ym1/2 have been linked to the increased tissue recruitment and activation of several of the innate and adaptive immune cells, such as Mϕs, DCs, NK cells, NK-T cells, B cells, and T cells, which cause visceral and brain tissue inflammation in *NPCA/B* disease [155,156].

*NPC1* and *NPC2* diseases are characterized by their impact on various organs, such as the liver, spleen, lung, lymph nodes, brain, and neurons, leading to symptoms such as hepatosplenomegaly, anemia, susceptibility to recurring infections, difficulty in walking and swallowing, progressive loss of hearing, and progressive development of dementia [26]. Elevated levels of C3, CR3, inactivated C3b (iC3b), complement receptor 4 (CR4), and mitogen-activated protein kinase kinase kinase 1 (MAP3K1) have been linked to liver damage in a mouse model of *NPC1* (Npc1^−/−^) [157,158]. Cerebellum and cerebellar Purkinje neurons, microglial cells, and CD68^+^ cells of Npc1^−/−^ mice have shown elevated levels of C1qa, C1qb, C1qc, C1r, C3, and C3aR [159,160,161]. The cerebella and cerebral cortex of Npc1^−/−^ mice have shown elevated levels of IFNα, IFNβ, TNFα, IL1α, IL1β, and the loss of Purkinje cells [157,161,162,163,164]. The plasma of patients with Niemann–Pick type C disease showed the presence of IgG and IgM antibodies to GM1, GM2, and GM3 [165]. The postmortem brain tissue (e.g., frontal cortex and cerebellum) from patients with NPC1 disease showed elevated levels of C3 and C4 [163]. The cerebrospinal fluid, frontal cortex, cerebellum, and fibroblasts from the patients with NPC1 have shown increased levels of TNFα, IL6, IL8, and IL17 [163,166]. 

Overall, these findings indicate that defects in the *NPC1/Npc1* gene lead to the excessive accumulation of various molecules, such as Sp1, GlycSph, Sm, and Ch, in the brain. This accumulation activates microglial cells and results in the generation of pro-inflammatory cytokines, including IFNα, IFNβ, TNFα, IL1α, and IL1β. In peripheral organs such as the liver, spleen, lung, and lymph nodes, *NPC1/Npc1* defects cause the activation of innate and adaptive immune cells, resulting in the production of GM1, GM2, and GM3-specific IgG and IgM autoantibodies and the generation of several pro-inflammatory cytokines, including IFNα, IFNβ, IL1α, IL1β, TNFα, IL6, IL8, and IL17.

The accumulation of such pro-inflammatory mediators may penetrate the blood–brain barrier, leading to neuronal cell death and the development of neurodegeneration and dementia in Niemann–Pick type C disease. These findings provide important insights into the disease mechanism and highlight multiple targets for the development of effective therapies to control central nervous system inflammation in Niemann–Pick type C disease. Further research in this area is crucial for the development of effective treatments for this devastating disorder.

## 7. Farber-Disease-Associated Neuroinflammation: Deciphering the Complex Interactions between Neurological and Immune Systems

Ultra-rare diseases are illnesses that affect <1 per 50,000–100,000 live births [4,167]. With this given definition, Farber disease is an ultra-rare autosomal recessive lysosomal disorder, which affects ~<1/1,000,000 live births [27]. Farber disease is caused by mutations in the *ASAH1 (in human)/Ash1 (in mouse)* gene and the resultant deficiency of the ACDase, EC 3.5.1.23 [168]. ACDase is a lipid hydrolase, which catalyzes the lysosomal degradation of Cer to Sph and free fatty acid [169]. Cer is important for performing several of cellular functions, such as apoptosis and inflammation [170,171]. *ASAH1/Asah1 defect* and the resulting excess tissue accumulation of Cer affect joint tissues, liver, heart, kidneys, lymph nodes, and central nervous system and are characterized by the symptoms of painful swelling of joints, arthritis, subcutaneous nodules, and hoarseness by laryngeal involvement, hepatosplenomegaly, nervous system dysfunction, and the increased lethargy and sleepiness [27,28,172]. 

Based on the excess tissue storage of Cer and its impact on the severity of the symptoms, Farber disease has been classified into types 1, 2, 3, 4, 5, 6, and 7. Types 1, 2, 3, 4, 5, and 6 Farber diseases are associated with the mutations in the *ASAH1* gene and the resulting deficiency of AC, whereas type 7 Farber disease is caused by the deficiency of the prosaposin, which is critical for regulating the function of lysosomal enzymes and the development of the neuroprotection [173]. Type 1 Farber disease exhibits lung involvement, neurological defects, and death at ~2 years of age. Type 4 Farber disease exhibits hepatosplenomegaly and death at ~2 years of age. Types 2, 3, 5, 6, and 7 Farber diseases show a less severe phenotype and survived up to adulthood [174,175]. The plasma and the Mϕs of the Asah1P361R/P361R Farber disease mouse model exhibited elevated levels of Cers, gangliosides, and their link to the higher levels of pro-inflammatory cytokines (e.g., IL1α, IL6, IL10, and IL12), chemokines (e.g., CCL2, CCL3, CCL12, CXCL1, CXCL9, and CXCL10), and the growth factor (e.g., VEGF) [176]. Plasma, lymph nodes, liver, kidney, lung, retina, hepatocytes, endothelial cells, epithelial cells, fibroblasts, sweat gland, and the Schwann cells of patients with Farber disease showed elevated levels of Cer and ganglioside [176,177,178,179,180,181]. Additionally, increased plasma levels of IL6, IL10, IL12, CCL2, CCL3, CXCL1, and VEGF and substantial central nervous system defects have been observed in patients with Farber disease [176,182]. 

The results of these studies suggest that the accumulation of Cer in the central and peripheral nervous systems can lead to the activation of microglial cells and excessive production of pro-inflammatory cytokines, particularly IL1α and IL6. This pro-inflammatory environment can contribute to the death of the central nervous system and peripheral nervous system cells, such as neurons and Schwann cells, and the development of neurodegeneration and cognitive defects in Farber disease. In addition to the effects in the central nervous system and peripheral nervous system, excess Cer accumulation in peripheral tissues can also lead to the recruitment of immune cells and the overproduction of pro-inflammatory cytokines, chemokines, and growth factors. This can result in damage to the blood–brain barrier and blood–nerve barrier, further exacerbating the effects of Cer accumulation in the central nervous system and peripheral nervous system. These findings provide important insights into the disease mechanisms underlying Farber disease and the role of Cer accumulation in the development of the central nervous system and peripheral nervous system inflammation. These findings also suggest potential targets for the development of new therapies to control the pro-inflammatory environment and improve the prognosis for patients with Farber disease.

## 8. Krabbe-Disease-Associated Neuroinflammation: Deciphering the Complex Interactions between Neurological and Immune Systems 

Krabbe disease is also known as galactosylceramide lipidosis or globoid cell leukodystrophy, which is also an ultra-rare disease that affects ~1/100,000–1/250,000 live births [183]. Krabbe disease is caused by the mutations in the *GALC (in human)/Galc (in mouse*) gene, which encodes an enzyme known as GALCERase (EC 3.2.1.46) critical for catabolism of GalCer, which is present in myelin, kidney, epithelial cells of the small intestine and the colon and responsible for the maintenance of proper structure and stability of myelin and differentiation of oligodendrocytes [57,184]. GALCERase is involved in the normal turnover of myelin by hydrolyzing GalCer, a major sphingolipid in the myelin composition [185]. *GALC*/*Galc* defects mainly lead to the deficiency of GALCERase and the progressive accumulation of GalCer and GalSph/psychosine that causes the complete disappearance of myelin in myelin-forming cells, i.e., oligodendrocytes in the central nervous system and the Schwann cells in the peripheral nervous system [39,186]. 

Krabbe disease is fatal and death occurs before the age of 5 in most cases with symptoms onset in early infancy [187]. Psychosine-induced in vitro and in vivo cell stimulation have shown increased production of pro-apoptotic factors, reactive astrocytic gliosis, and infiltration of the multinucleated Mϕs (globoid cells) which propagate the disease by causing the death of oligodendrocytes, Schwann cells, and neurons [188,189,190,191]. The GALCERase defect and the resultant excess tissue accumulation of GalCer have not been observed in Krabbe disease tissues. However, the excess accumulation of the minor substrate of GALCERase, i.e., GalSph, triggers extensive demyelination in Krabbe disease [186]. Krabbe disease is classified into infantile, juvenile, and adult-onset forms. More than 85% of patients with Krabbe disease exhibit the rapidly progressive infantile-onset form of the disease, which leads to death by 2 years of age [192]. Children affected with infantile Krabbe disease display worsening mental and motor skills, muscle weakness, hypertonia, spasticity, myoclonic seizures, irritability, unexplained fever, deafness, blindness, paralysis, difficulty in swallowing, and weight loss [29,30]. 

The Twither (Twi) mouse model of Krabbe disease has shown elevated brain levels of GalSph. The brain and their different regions (e.g., brain stem, cerebellum, cortex, forebrain, and hindbrain) of these mice have shown activated subsets of CD68^+^ Iba1^+^ microglial cells and GFAP^+^ astrocytes, and the pro-inflammatory cytokines (e.g., IFNγ, TNFα, IL1α, IL1β, IL6, and IL10) and chemokines (e.g., CCL2, CCL3, CCL11, CCL12, CXCL1, CXCL9, and CXCL10) [193]. Similarly, the brains of the humanized mouse model and patients with globoid cell leukodystrophy have shown the excess accumulation of GalSph and they are linked to the presence of activated subsets of CD68^+^ Iba1^+^ microglial cells and GFAP^+^ astrocytes, and the pro-inflammatory cytokines [186,194]. 

These research findings indicate that defects in the *GALC*/*Galc* gene cause an excess accumulation of GalCer and/or GalSph in both the central nervous system and peripheral nervous system. This accumulation triggers the activation of microglial cells and Schwann cells, which leads to the massive production of pro-inflammatory cytokines, such as IFNγ, TNFα, IL1α, IL1β, IL6, and IL10, as well as chemokines, such as CCL2, CCL3, CCL11, CCL12, CXCL1, CXCL9, and CXCL10. Similarly, the accumulation of GalCer and/or GalSph in peripheral tissues also results in the activation of macrophages and the overproduction of pro-inflammatory cytokines and chemokines in Krabbe disease. The development of such a pro-inflammatory environment in the central nervous system, peripheral nervous system, and peripheral organs can disrupt the blood–brain barrier/blood–nerve barrier, causing neuronal, oligodendrocyte, and Schwann cell death. This can lead to severe neurological symptoms, such as developmental delay, seizures, loss of motor function, and cognitive defects in Krabbe disease. These insights provide a better understanding of the disease mechanism and the critical targets that can be used to develop effective therapies for Krabbe disease.

## 9. Wolman Disease—Associated Neuroinflammation: Deciphering the Complex Interactions between Neurological and Immune Systems 

Wolman disease is also known as cholesteryl ester storage disease, which affects ~1/350,000 live births [195]. Wolman disease is an autosomal recessive lysosomal storage disorder caused by mutations in *LIPA (in human)/Lipa (in mouse*) gene that encodes the enzyme called LAL (EC 3.1.1.13) responsible for the hydrolysis of CEs and TGs [196]. This enzyme contributes to the homoeostatic control of plasma lipoprotein levels and to the prevention of cellular lipid overload in the liver, spleen, and their Mϕs [197]. The Cer and TGs are involved in various cellular processes, which regulate lipid–protein interactions and the cell signaling events critical for the proper function of the organs [198,199,200,201]. However, *LIPA*/*Lipa* defect and resulting deficiency of LAL cause progressive lysosomal accumulation of CEs and TGs, which can affect multiple organs, such as the liver, spleen, adrenal gland, and intestine, as well as a variety of cells including neuronal stem cells, hepatocytes, MOs, and Mϕs. 

The Wolman disease symptoms include mental deterioration, enlarged liver and spleen, distended abdomen, gastrointestinal problems, jaundice, anemia, vomiting, calcium deposition in adrenal glands, and hypothyroidism [31,32,202]. Clinically, LAL deficiency results in two major phenotypes, i.e., infantile-onset Wolman disease and later-onset cholesteryl ester disease, and correlates with higher residual LAL activity relative to Wolman disease. Patients with cholesteryl ester disease presentation occur later in life with hepatomegaly, hyperlipoproteinemia, and premature atherosclerosis. LAL-prone (LAL^−/−^) and the conditional transgenic mouse model of Wolman disease have shown the elevated tissues (e.g., liver, spleen, and small intestine) and their cells (e.g., Mϕs) level of CE and the TGs [197,203]. Additionally, plasma, liver, and hepatocytes of LAL-prone (LAL^−/−^) and the conditional transgenic mouse model of Wolman disease have shown increased production of pro-inflammatory cytokines (e.g., IFNγ, TNFα, IL2, IL4, and IL6), chemokines (e.g., CCL2, CCL3, CCL4, CCL5, and CXCL10), and growth factors, i.e., MCSF and GMCSF [203]. The liver, hepatocytes, adrenal glands, intestine, and cells of the MO–Mϕs system of the patients with Wolman disease have shown elevated liver, adrenal glands, intestines, and the Mϕs–monocyte level of CEs and the TGs [204].

The findings of these studies suggest that *LIPA*/*Lipa* gene defects and the resulting excess tissue accumulation of CEs and/or TGs lead to microglial cells and Mϕs activation and cause increased production of pro-inflammatory cytokines in the brain. Similarly, the accumulation of CEs and/or TGs in peripheral tissues (e.g., liver, spleen, and small intestine) causes the overproduction of chemokines and growth factors, leading to tissue recruitment of T cells, MO-differentiated Mϕs, and DCs, and causes the increased production of pro-inflammatory cytokines. The development of such pro-inflammatory environment in the central nervous system and peripheral organs affects the blood–brain barrier and causes neuronal cell death, resulting in the development of neurodegeneration in Wolman disease. These findings provide valuable insights into the disease mechanism and potential therapeutic targets for controlling inflammation in both the central nervous system and peripheral organs in Wolman disease.

## 10. Discussion

This research highlights the connection between genetic defects and inflammatory conditions, specifically the accumulation of excess tissue substrates. This accumulation leads to the activation of immune and neurological cells, resulting in the production of various pro-inflammatory substances, including cytokines, chemokines, complement components, autoantibodies, and growth factors. The end result is tissue destruction in lysosomal storage disease. Studies have demonstrated a comprehensive understanding of current approaches (e.g., gene therapy, enzyme replacement therapy, substrate reduction therapy, hematopoietic stem cell transplantation, pharmacological chaperone therapy, and the proteostasis regulators) for the management of complications of lysosomal storage diseases [205,206,207,208,209]. However, the current treatment experiences difficulty targeting the pathologic organs, including the CNS, PNS, retina, and skeletal system. Additionally, these treatments are very expensive, do not provide a complete cure, and risk of developing cancer, graft rejection, and immune inflammation leading to potentially life-threatening reactions [205,206,207,208,210,211,212,213,214,215,216]. Overall, further research is needed to identify new therapeutic targets that can resolve the issues, where current therapies are not effective controlling the immune inflammation that leads to the tissue destruction in lysosomal storage diseases.

One potential avenue for developing new treatments is the identification of pro-inflammatory mediators in the CSF/bloodstream, which could serve as therapeutic biomarkers for these diseases. Identification and monitoring of these biomarkers may be able to develop targeted therapies that can more effectively combat the underlying inflammation and tissue destruction associated with lysosomal storage diseases. Overall, this research underscores the complex relationship between genetics, inflammation, and disease and highlights the ongoing need for further study and innovation in the field of lysosomal storage disease research.

Certain inflammatory conditions cause accelerated migration of immunological cell precursors out of the bone marrow and into the circulation and from the bone marrow and circulation to the sites of inflammation for the generation of pro-inflammatory cytokines that lead to tissue destruction [217,218,219,220,221,222,223,224,225,226,227,228]. A similar condition is thought to occur in lysosomal storage disease due to the genetic defects and the resultant excess peripheral organs (e.g., liver, spleen, lung, heart, kidneys, and lymph nodes) accumulation of distinct substrates (e.g., GC in Gaucher, Gb3 and Lyso Gb3 in Fabry, GM1 in GM1gangliosidosis, GM2 in Tay–Sachs and Sandhoff, Sph, GlycSph, Sm, and Ch in Niemann–Pick type C, Cer in Farber, GalSph in Krabbe, and CEs and TGs in Wolman diseases), which lead to the cellular activation and increased production of the growth factors, such as MCSF, GCSF, and GMCSF, which mobilize hematopoietic progenitors into the peripheral circulation [229,230,231,232]. This is supported here by the elevated level of growth factors (e.g., MCSF, GCSF, and GMCSF), chemoattractants (C3a, C5a, CCL2, CCL3, CCL4, CCL5, CCL10, CCL11, CCL12, CXCL1, CXCL9, CXCL10, CXCL11, and CXCL13), and the increased presence of MOs, granulocytes, MOs-differentiated Mϕs and DCs, T cells, NK cells, NKT cells, antibodies producing plasma B cells in circulation and the peripheral organs of the different lysosomal storage diseases [6,7,11,36,76,112,113,157,158,165,176,186,194,203,204,222,233,234,235,236,237,238,239,240,241,242,243,244]. The increased migration of several of such pro-inflammatory mediators into circulation may serve as a new set of biomarkers for diagnosing lysosomal storage diseases and evaluating the effectiveness of novel medications in clinical trials.

The blood–brain barrier and blood–nerve barrier are highly selective semipermeable barriers that control the exchange between the blood and the nerve tissues. The blood–brain barrier is made up of specialized microvascular endothelial cells with tight junctions (TJs) proteins composed of transmembrane and peripheral proteins, such as claudin 5, 3, and 12, while the blood–nerve barrier has a similar structure but is more permeable than the blood–brain barrier [245,246,247,248,249,250,251,252,253]. The blood–brain barrier separates the central nervous system from the peripheral tissues and controls the transfer of nutrients and cells from the blood to the brain and from the brain to the blood and protects the central nervous system [251,254,255,256,257,258]. These are important for restricting the paracellular flow of ions and molecules into the endoneurial milieu [252,253]. The blood–nerve barrier prevents access to cellular and humoral factors and cells from the circulation [252,253].

Increased neural and visceral tissue expression of several of the inflammatory mediators, such as matrix metalloproteinases 2 (MMP2) and MMP-9 [259,260,261], oxidative stress [262], pro-inflammatory cytokines, such as IFNγ [263], IL1β [264,265], TNFα [266,267], IL6 [266], and IL17 [268,269,270], chemokines, i.e., CCL2 [271,272], CCL4 [273], CCL5 [274], CXCL10 [275], autoantibodies, C5a [276,277,278,279,280,281,282,283,284,285,286,287], and their effector functions are critical for blood–brain barrier damage, death of the central nervous system cells (e.g., astrocytes, microglial cells, and oligodendrocytes), neuronal dysfunction, neurodegeneration, and cognitive impairment in several mouse models and human brain diseases, i.e., intracerebral hemorrhage, traumatic brain injury, systemic lupus erythematosus, myasthenia gravis, amyotrophic lateral sclerosis, neuromyelitis optica spectrum, ischemic stroke, epilepsy, major depression, mood disorders, psychosis, autism spectrum disorder, chronic sleep disorder, Alzheimer, Parkinson, and Huntington’s illnesses [251,262,266,275,276,277,278,279,280,281,282,283,284,285,286,287,288,289,290,291,292,293,294,295,296,297,298,299,300,301,302,303,304,305,306,307,308,309,310,311,312,313]. 

Upon axonal injury or acute demyelination, the blood–nerve barrier becomes leaky and circulating cellular and humoral immune components such as complement activation products and the pro-inflammatory cytokines enter the nerve in several peripheral neuropathies [314,315,316,317,318,319,320,321,322,323]. Elevated levels of IL17-producing cells, IL17 cytokine, and MMPs have been linked to the marked reduction in the level of TJs proteins of the blood–brain barrier in a mouse model of lysosomal storage disease known as infantile neuronal ceroid lipofuscinosis or the Batten disease [324]. It is possible that the integration of immune cells and the resulting generation of pro-inflammatory cytokines in the periphery can alter the integrity of both the blood–brain barrier and blood–nerve barrier. This can create a situation in which pro-inflammatory cytokines are able to enter the blood–brain barrier and blood–nerve barrier, potentially leading to damage in the central nervous system and peripheral nervous system. This can result in neuronal cell loss and the development of neurodegeneration in lysosomal storage disease.

The precise mechanisms of alterations in the blood–brain barrier and blood–nerve barrier in lysosomal storage disease are not yet clear and require further investigation. However, current findings indicate that genetic defects and subsequent excess accumulation of specific substrates in the central nervous system and peripheral nervous system tissues activate microglial cells, astrocytes, oligodendrocytes, and Schwann cells. This leads to a massive generation of pro-inflammatory cytokines, such as IFNα, IFNβ, IFNγ, TNFα, IL1α, IL1β, and IL6, resulting in cellular death in the central nervous system and peripheral nervous system cells. Similarly, the genetic defect and excess accumulation of substrates in peripheral tissues, such as the liver, spleen, lung, lymph nodes, and intestine, lead to the overproduction of growth factors and chemokines, triggering mobilization of bone marrow and activation of both innate and adaptive immune cells. This results in a massive generation of various pro-inflammatory cytokines, which penetrate the blood–brain barrier and blood–nerve barrier and cause the death of the central nervous system and peripheral nervous system cells and the development of neurodegeneration in lysosomal storage diseases (Figure 1A–J). 

The activation of mTOR (mechanistic target of rapamycin), LAT (linker for activation of T cells), PI3K (phosphoinositide 3-kinase), AKT (protein kinase B), JNK (c-Jun N-terminal kinases), RIP3K (receptor-interacting protein kinase 3), ERK (extracellular signal-regulated kinase), and p38-MAPK (p38 mitogen-activated protein kinases) signaling causes the massive production of IFN-α, IFN-β, IFN-γ, IL1 α, IL1 β, TNF α, IL6, IL12, IL17, and C5a [325,326,327,328,329,330,331,332,333,334,335,336,337]. Studies have demonstrated the involvement of several of the signaling cascades in lysosomal storage diseases, particularly in the context of disease pathogenesis and potential therapeutic targets. In particular, the mTOR, LAT, PI3K, AKT, JNK, RIPK3, ERK, MAP3K1, and p38MAP signaling has been implicated in the regulation of lysosomal biogenesis, autophagy, and complement activation, which are key cellular processes involved in the tissue damage in GM2 gangliosidosis, mucopolysaccharidosis, Niemann–Pick type C, Gaucher, and Fabry diseases [36,157,158,338,339,340,341,342,343,344,345,346,347,348,349]. These findings suggest that inhibition of mTOR, LAT, PI3K, AKT, JNK, RIP3K, ERK, MAP3K1, and p38MAPK- signaling can stop the neuroinflammation in lysosomal storage diseases. However, while the study of signaling cascades in lysosomal storage diseases is a promising area of research, many questions remain unanswered. It is not yet clear how different signaling pathways interact with one another and how their dysregulation contributes to disease pathogenesis. Additionally, the potential side effects of targeting these pathways in vivo remain to be fully understood. The study of signaling cascades in lysosomal storage disorders represents a promising avenue for the development of novel therapies for these disturbing diseases. 

Although current research has shed some light on the potential interplay between signaling pathways and pro-inflammatory mediators in lysosomal storage diseases, it is crucial to recognize the complexity of this relationship. There are still many unanswered questions, including how lysosomal defects, enzyme and protein alterations, lipid dysregulation, and autoimmune responses contribute to disease progression. To better understand the mechanistic connection between signaling pathways and pro-inflammatory mediators in lysosomal storage diseases, both human and animal studies are necessary. Such studies will help identify the role of immune responses in the pathogenesis of these diseases and uncover the most effective therapeutic targets. 

## Figures and Tables

**Figure 1 biomedicines-11-01067-f001:**
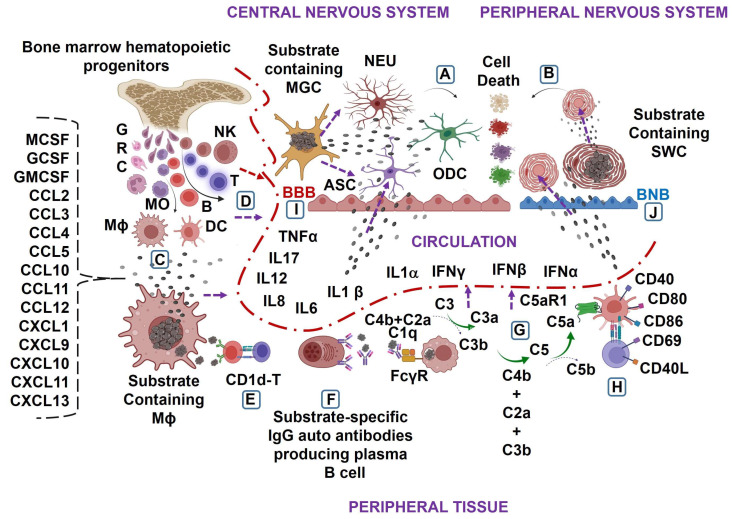
Model of neuroimmune inflammation induced by genetic defects and excess substrate accumulation in lysosomal storage diseases. Genetic defects can lead to the excessive accumulation of substrates associated with lysosomal storage diseases. In the central nervous system, microglial cells (MGCs) and, in the peripheral nervous system, Schwann cells (SWCs) are particularly affected. This accumulation of substrates can cause the local production of pro-inflammatory cytokines, including interferon alfa (IFNα), interferon beta (IFNβ), interferon gamma (IFNγ), tumor necrosis factor alfa (TNFα), interleukin (IL)1α, IL1β, and IL6, resulting in the death of neurons (NEU), astrocytes (ASC), oligodendrocytes (ODC), and SWCs in lysosomal storage diseases (**A**,**B**). Similarly, the excess accumulation and continuous release of substrates from macrophages (Mϕ) in peripheral tissues can cause the massive generation of growth factors, such as Mϕ-colony-stimulating factor (MCSF), granulocyte-colony-stimulating factor (GCSF), and granulocyte-Mϕ colony-stimulating factor (GMCSF), C-C motif ligand chemokines (e.g., CCL2, CCL3, CCL4, CCL5, CCL10, CCL11, and CCL12), and C-X-C motif ligand chemokines (e.g., CXCL1, CXCL9, CXCL10, CXCL11, and CXCL13), leading to the bone marrow mobilization of immunological cell precursors and the excess peripheral tissue recruitment of T, B, and natural killer (NK) cells, granulocytes (GRC), monocyte (MO), MO-differentiated Mϕ and dendritic cell (DC), and massive circulatory release of pro-inflammatory cytokines in lysosomal storage diseases (**C**,**D**). Furthermore, in lysosomal storage diseases, the excessive accumulation and continuous release of substrates from tissue Mϕ result in cellular activation, plasma B-cell formation, and the massive production of substrate-specific IgG autoantibodies (**E**,**F**). These IgG autoantibodies react with activating Fcγ receptors present on Mϕ and causes the formation of the enzymes C3 convertase, i.e., C4b + C2a = *C4b2a* and/or C5 convertase, i.e., C3b + C3b + C2a + C4b + C3b + C3b = *C4b2a (C3b) n*. The activation of the C3 convertase results in the downstream cleavage of C3 into C3a and C3b, whereas the activation of the C5 convertase leads to the production of C5a (**G**). The interaction of C5a with the C5aR1 receptor upregulates the expression of costimulatory molecules (e.g., CD40, CD80, and CD86) on antigen presenting cells such as DCs and stimulatory molecules (e.g., CD69 and CD40L) on CD4^+^ T cells, eventually leading to the massive generation of pro-inflammatory cytokines in lysosomal storage diseases (**H**). These pro-inflammatory cytokines affect the blood-brain barrier (BBB) and blood-nerve barrier (BNB), promoting their interaction with the central nervous and peripheral nervous system residential cells, i.e., NEU, ASC, ODC and SWC, and cause their death, leading to the development of neurodegeneration in lysosomal storage diseases (**I**,**J**).

**Table 1 biomedicines-11-01067-t001:** The clinical symptoms of lysosomal storage diseases caused by genetic defects and enzyme deficiencies.

Lysosomal Storage Disease	Gene Defects	Enzyme or Protein Deficiency	Substrate Accumulation	Affected Organs and Cells	Clinical Symptoms	Reference
Gaucher disease	*GBA1*(in human)/ *Gba1*(in mouse)	GCase	GC, GS	Central nervous system tissues, microglial cells, neurons, liver, spleen, lung, kidney MO, Mϕ, DC	Seizure, abnormal eye movements, developmental delay, hepatosplenomegaly, pulmonary inflammation, skeletal weakness, hypergamma globulinemia, B-cell malignancies, anemia, and thrombocytopenia	[5,6,7]
Fabry disease	*GLA* (in human)/*Gla* (in mouse)	α Gal A	Gb3 and lyso-Gb3	Central nervous system tissue, and their cells, liver, spleen, kidney, blood vessel walls, renal epithelial cell, endothelial cell, pericyte podocytes, tubular cell of the loops of Henle and the thick ascending limb of the distal tubule, vascular smooth muscle cell, and cardiomyocyte	Stroke, burning pain in the arms and legs, cardiomegaly, renal failure, gastrointestinal difficulties, decreased sweating, fever, and angiokeratomas	[8,9,10,11,12,13,14,15,16]
GM1 gangliosidosis	*GLB1* (in Human)/*Glb1*(in mouse)	β gal	GM1	Central and peripheral nervous system tissue, spleen, bone, and muscle	Dementia, seizures, hyperekplexia, abnormal gait, ataxia, stuttering, apraxia, dysarthria, coarse facies, muscle atrophy, dystonia, angiokeratoma, skeletal irregularities, joint stiffness, abdominal distension, muscle weakness, and hepatosplenomegaly	[17,18,19]
GM2 gangliosidosis: Tay–Sachs disease	*HEXA* (in human)/*Hexa* (in mouse)	α subunit of the β hex	GM2	Central nervous system tissue	Dementia, paralysis, seizures decreased eye contact, increased startle response to noise, deafness, dysphagia, blindness, cherry-red spots in the retina	[20,21,22,23]
GM2 gangliosidosis: Sandhoff disease	*HEXB* (in human)/*Hexb* (in mouse)	β subunit of the β hex	GM2	Central nervous system tissues, liver, spleen, lung, and heart	Motor weakness, early blindness, startle response to sound, spasticity, myoclonus, seizures, macrocephaly, cherry red spots in the eye, doll-like facies, frequent respiratory infections, heart murmurs, and hepatosplenomegaly	[20,21,22,23,24,25]
Niemann–Pick type C disease	*NPC1*/*NPC2*(in human)/*Npc1*/*Npc2*(in mouse)	NPC1 and NPC2	Sph, GlycSph, Sm, and Ch	Peripheral nervous system tissue, nerve cell, liver, spleen, lymph node, hard lump under the skin	Progressive dementia, difficulty in walking, dysphagia, progressive loss of hearing, hepatosplenomegaly, anemia, and susceptible to recurring infection	[26]
Farber disease	*ASAH1* (in human/*Asah1*(in mouse)	ACDase	Cer	Central nervous system tissue, liver, heart, kidney, lymph node, and joints	Increased sleep and tiredness, dysphagia, dysphonia, arthrogryposis, vomiting, and arthritis	[27,28]
Krabbe disease	*GALC* (in human/*Galc* (in mouse)	GALCERase	GalCer and GalSph or psychosine	Myelin sheath	Dyspraxia, myoclonic seizures, deafness, blindness, paralysis, dysphagia, muscle weakness, hypertonia, spasticity, and weight loss	[29,30]
Wolman disease	*LIPA* (in human)/*Lipa* (in mouse)	LAL	CEs and TGs	Liver, spleen, blood, lymph, adrenal gland, MO, and Mϕ	Mental deterioration, Hepatomegaly, abdominal distension, gastrointestinal problems, jaundice, anemia, vomiting, and calcium deposits in the adrenal glands	[31,32]

*GBA1*/*Gba1*, glucosylceramidase beta 1; GCase, β-glucocerebrosidase; GC, glucosylceramide; GS, glucosphingosine; *GLA*/*Gla*, galactosidase alpha; α-Gal A, alpha-galactosidase A; Gb3, globotriaosylceramide; lyso-Gb3, globotriaosylsphingosine; *GLB1*/*Glb1*, galactosidase beta 1; β gal, β galactosidase; GM1, ganglioside GM1; GM2, ganglioside GM2; *HEXA*/*Hexa*, hexosaminidase A; *HEXB*/*Hexb*, hexosaminidase B; β hex, β hexosaminidase; *NPC1*/*Npc1*, Niemann–Pick type C1; *NPC2*/*Npc2*, Niemann–Pick type C2; Sph, sphingosine; GlycSph, glycosphingolipid; Sm, sphingomyelin; Ch, cholesterol; *ASAH1*/*Asah1*, N-acylsphingosine amidohydrolase 1; ACDase, acid ceramidase; Cer, ceramide; *GALC*/*Galc*, galactosylceramidase; GALCERase, galactosylceramidase; GalCer, galactosyl ceramide; GalSph, galactosylsphingosine; *LIPA*/*Lipa*, lysosomal acid lipase; LAL, lysosomal acid lipase; CEs, cholesteryl esters; TGs, triglycerides; MO, monocyte; Mϕ, macrophage; DC, dendritic cell.

## Data Availability

Not applicable.

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
