# Peer review of "Exploring Pro-Inflammatory Immunological Mediators: Unraveling the Mechanisms of Neuroinflammation in Lysosomal Storage Diseases"

_biomedicines, 2023, doi:10.3390/biomedicines11041067_

Round 1
Reviewer 1 Report
The manuscript submitted by Pandey summarize the contribution of inflammatory pathways across the lysosomal storage diseases reported, focusing on pro-inflammatory mediators found in human and murine models, as responsible for the death of peripheral and central nervous system cells, and as possible biomarkers. The review topic is very topical and of great interest since neuroinflammation seems to be a widely shared mechanism in pathologies involving the central and peripheral nervous system and a deep understanding of it is needed.
The review reports findings on patients and murine models of pathologies underling a connection between genetic mutations responsible for the onset of the diseases and the neuroinflammation. Although the paper is well articulated and clear, it may be useful to report recent research in which the contribution of neuroinflammation in the onset and progression of the diseases has been demonstrated by modulating the expression of the inflammatory markers mentioned in the review; e.g. silencing/downregulating their expression; inhibiting their activity; highlighted their downstream pathways, thus proving the author’s hypothesis about the immune system activation and the involvement of neuroinflammation in lysosomal storage diseases.
Also, a summary of what is common and what is different in terms of inflammatory pathways between the different pathologies might be useful to have a clear and immediate overview of mechanisms of neuroinflammation in lysosomal storage diseases.
Author Response
I am deeply grateful to the reviewer for dedicating their time to reviewing this work and providing such valuable feedback. I have taken into consideration your insightful suggestions, I have included several recent research findings that highlight the significance of signaling cascades in lysosomal storage diseases, particularly in terms of disease pathogenesis and potential therapeutic targets, such as mammalian target of rapamycin (mTOR), linker for activation of T cells (LAT), phosphoinositide 3-kinase (PI3K), protein kinase B (AKT), Jun N-terminal kinases (JNK), Receptor-interacting protein kinase-3 (RIPK3), and p38 mitogen-activated protein kinase (p38MAP-Kinase).
Additionally, suggested summary of the involvement of inflammatory components have been included under the section of each lysosomal storage disease.
Reviewer 2 Report
Dear Author,
congratulations for a great work and for the innovative informations on LSD. We suggest to put in evidence the pathogenetic mechanisms of LSD described in the paper Lysosomal storage diseases: from pathophysiology to therapy by Parenti G, Andria G, Ballabio A. Annu Rev Med. 2015;66:471-86.
We appreciate the figures and the table, in a well structured and detailed paper that involve salso the peripheral nervous system and the blood-nerve-barrier.
In the Fabry description could you specify if you referred to the classic form?
Occasionally the paper is very intrigate and complex, we suggest to simplify the text for a better lecture.
Author Response
I would like to express my gratitude to reviewer for taking the time to review this work and providing with such insightful comments. I have taken your suggestions into consideration and made the necessary changes to improve the quality of our research. I have included the suggested reference of Parenti et al (PMID: 25587658) in our manuscript.
Regarding the Fabry description, this paper has provided valuable insights into the work done in individuals with the classic and late-onset forms of the condition who may experience organ damage, primarily in the heart, kidney, and brain. However, it is important to note that the percentage of patients who experience severe problems in these organs is much lower. This is supported by the cited references; PMID: 23452955; PMID: 22085605, and PMID: 25619383. All these information has been included and entire paper has been simplified for a better construing.
Reviewer 3 Report
In this review, author summarized the genetic defects associated with lysosomal storage diseases and their impact on the induction of neuroinflammation.
Author claimed that the pro-inflammatory mediators in the bloodstream may be used as biomarkers for diagnosing the lysosomal storage disease or evaluating the effectiveness of new medications in clinical trials. Due to the rareness of these diseases and multiple reasons that can cause production of pro-inflammatory mediators, it will be inaccurate to say that these pro-inflammatory mediators can be used as diagnostic biomarkers for these diseases. While the pro-inflammatory mediators panel could be used as therapeutic biomarkers for these diseases.
Author used so many references for some information such as lines 536 and 555, please keep only references related to the stated information.
Conclusions
Author must restate the conclusion section to include his explanation and opinion about what he summarized in the whole review not repeating the information again.
Figure 1. Please put circles around letters A-J to be easier to see the different compartments of figure.
Review needs minor English editing.
Author Response
I would like to express my sincere gratitude for the valuable feedback that reviewer provided on my paper. Your comments were highly appreciated and have been instrumental in improving the quality of my work.
I have taken your suggestions into consideration and have made the necessary changes to better describe the role of pro-inflammatory mediator’s panel and their potential use as therapeutic biomarkers for the lysosomal storage disease discussed in my paper. These modifications can be found in the first two paragraphs of the discussion section.
Furthermore, I have also incorporated your additional suggestion correcting the Figure, minor English editing, and to restate the conclusion section and include my own opinion and explanation about the findings summarized in the entire review. As a result, I have extensively revised the entire paper to better reflect my insights on the topic.
Once again, I sincerely thank you for your time and effort in reviewing my paper. Your valuable input has been immensely helpful in enhancing the overall quality of my work.